# Replicability of bulk RNA-Seq differential expression and enrichment analysis results for small cohort sizes

Peter Methys Degen[iD][1,2,3]*, Matúš Medo[iD][1,2]

**1** Department for BioMedical Research, Radiation Oncology, University of Bern, Bern, Switzerland, **2** Department of Radiation Oncology, Inselspital Bern University Hospital, Bern, Switzerland, **3** Graduate School for Cellular and Biomedical Sciences, University of Bern, Bern, Switzerland

\* peter.methys@proton.me

**Data availability statement:** The GATB, GIPF, and HSPL data sets supporting the conclusions of this article are publicly available from the

## Abstract

The high-dimensional and heterogeneous nature of transcriptomics data from RNA sequencing (RNA-Seq) experiments poses a challenge to routine downstream analysis steps, such as differential expression analysis and enrichment analysis. Additionally, due to practical and financial constraints, RNA-Seq experiments are often limited to a small number of biological replicates. In light of recent studies on the low replicability of preclinical cancer research, it is essential to understand how the combination of population heterogeneity and underpowered cohort sizes affects the replicability of RNA-Seq research. Using 18'000 subsampled RNA-Seq experiments based on real gene expression data from 18 different data sets, we find that differential expression and enrichment analysis results from underpowered experiments are unlikely to replicate well. However, low replicability does not necessarily imply low precision of results, as data sets exhibit a wide range of possible outcomes. In fact, 10 out of 18 data sets achieve high median precision despite low recall and replicability for cohorts with more than five replicates. To assist researchers constrained by small cohort sizes in estimating the expected performance regime of their data sets, we provide a simple bootstrapping procedure that correlates strongly with the observed replicability and precision metrics. We conclude with practical recommendations to alleviate problems with underpowered RNA-Seq studies.

## Author summary

Transcriptomics data from RNA sequencing (RNA-Seq) experiments are complex and challenging to analyze due to their high dimensionality and variability. These experiments often involve limited biological replicates due to practical and financial constraints. Recent concerns about the replicability of cancer research highlight the need to explore how this combination of limited cohort sizes and population heterogeneity impacts the reliability of RNA-Seq studies. To investigate these issues, we conducted

GEO (https://www.ncbi.nlm.nih.gov/geo/). GEO accession numbers of studied data sets are GSE107994, GSE150910, and GSE247382, respectively. The SNF2 yeast data was downloaded from a third-party GitHub repository (https://github.com/ Morris-Research-Group/bayexpress). The TCGA data sets are publicly available from the GDC (https://portal.gdc.cancer.gov/). A persistent Git repository with Python scripts and notebooks for downloading the TCGA data and performing the analysis is available on Zenodo (https://doi.org/10.5281/zenodo.8333519). The repository also includes processed (aggregated) data sets which we used to generate the figures, as well as lists of ground truth DEGs and enriched terms for each data set. A standalone repository to perform bootstrapped analyses is available via GitHub (https://github.com/pdegen/BootstrapSeq).

**Funding:** This research project was supported by a grant from the Werner und Hedy Berger-Janser Foundation for cancer research (https://www.krebskrankheiten.ch/). The grant was awarded to MM. The funders had no role in study design, data collection and analysis, decision to publish, or preparation of the manuscript.

**Competing interests:** The authors have declared that no competing interests exist.

18'000 subsampled RNA-Seq experiments based on real gene expression data from 18 different data sets. We performed differential expression and enrichment analyses for each experiment to obtain significant genes and gene sets. We show that experiments with small cohort sizes tend to produce results that can be difficult to replicate. We further found that while underpowered experiments with few replicates indeed lead to little-replicable results, this does not mean that the results are necessarily wrong. Depending on the characteristics of the data set, the results may contain a large or small number of false positives. To help researchers with limited replication numbers estimate which is the case for their data sets, we demonstrate a simple resampling procedure to predict whether the analysis results are prone to false positives.

## Introduction

The rapidly increasing availability of large and highly heterogeneous omics data from high-throughput sequencing technologies has stimulated the development of appropriate statistical methods [1]. Differential expression analysis, the problem of detecting systematic differences in the expression levels of genomic features between experimental conditions (e.g., normal tissue versus tumor tissue), is a key problem in this field [2–4]. The term "genomic features" here can refer to genes, exons, transcripts, or any other genomic region of interest; we shall henceforth use the umbrella term "gene" for simplicity of notation. Gene expression levels are typically quantified using read counts obtained from next-generation sequencing technologies such as RNA-Seq [5]. Due to various sources of biological and technical variability, statistical hypothesis tests are needed to determine the significance of any observed difference in the read counts. Genes that pass a significance threshold after correcting for multiple hypothesis testing are designated as differentially expressed genes (DEGs). They can be used for further downstream analysis such as enrichment analysis [6] or independent scrutiny in subsequent wet lab experiments.

The statistical power of RNA-Seq experiments naturally increases with the number of biological replicates. However, a review of the available literature suggests that actual cohort sizes often fall short of the recommended minimum cohort sizes. For example, Schurch et al. [7] estimated that at least six biological replicates per condition are necessary for robust detection of DEGs, increasing to at least twelve replicates when it is important to identify the majority of DEGs for all fold changes. Lamarre et al. [8] argued that the optimal FDR threshold for a given replication number $n$ is $2^{-n}$, which implies five to seven replicates for typical thresholds of 0.05 and 0.01. Bacarella et al. [9] cautioned against using fewer than seven replicates per group, reporting high heterogeneity between the analysis results depending on the choice of the differential expression analysis tool. Ching et al. [10] estimated optimal cohort sizes for a given budget constraint, taking into account the trade-off between cohort size and sequencing depth. While emphasizing that the relationship between cohort size and statistical power is highly dependent on the data set, their results suggest that around ten replicates are needed to achieve $\gtrsim 80\%$ statistical power.

Despite this body of research warning against relying on insufficient replication numbers, three replicates per condition remains a commonly used cohort size, and many RNA-Seq experiments employ fewer than five replicates [7]. A survey by Baccarella et al. [9] reports that about 50% of 100 randomly selected RNA-Seq experiments with human samples fall at or below six replicates per condition, with this ratio growing to 90% for non-human samples. This tendency toward small cohorts is due to considerable financial and practical constraints that inhibit the acquisition of large cohorts for RNA-Seq experiments, as including

more patients in a study requires substantial time and effort, especially for rare disease types. More generally, a study by Dumas-Mallet et al. [11] suggests that as much as half of biomedical studies have statistical power in the 0–20% rage, well below the conventional standard of 80%. Button et al. [12] estimated the average power for neuroscience studies to be 21%. Modeling using optimization theory suggests that current incentive structures in science favor researchers who publish novel results from underpowered studies, resulting in half of all studies supporting erroneous conclusions [13]. In light of the prevalence of underpowered research, there is an urgent need for further investigation into the potentially detrimental effects of low-powered RNA-Seq experiments. Unfortunately, recent literature on this topic is limited.

One recent study was conducted by Cui et al. [14], who subsampled RNA-Seq data from The Cancer Genome Atlas (TCGA) [15] and calculated the overlap of DEGs among the subsampled cohorts. Noting the low overlap of results for small cohort sizes, the authors recommend using at least ten replicates per condition and interpreting low-powered studies with caution. Another study based on the TCGA data was conducted by Wang et al. [16], whose primary concern was the comparison of different metrics for evaluating replicability. The authors report significantly heterogeneous results depending on the chosen replicability metric and the studied cancer type.

More generally, Ioannidis [17] proposed a simple statistical model for high-throughput discovery-oriented research, of which RNA-Seq is a prime example. This model can be used to demonstrate potentially high rates of false positive results. Although the author's claim that "most published research findings are false" has been the subject of considerable debate [18, 19], it indeed appears to be the case that certain fields such as preclinical cancer biology are struggling with a high prevalence of research with replication problems [20,21]. Errington et al. [22] recently conducted a large-scale replication project that attempted to replicate 158 effects from 50 experiments in 23 high-impact papers in preclinical cancer research, achieving a success rate of 46%. Furthermore, the authors found that 92% of the replicated effect sizes were smaller than in the original study.

Motivated by the described issues, our goal is to investigate the replicability and reliability of RNA-Seq analysis results obtained from small cohort sizes. Compared to the studies by Cui et al. [14] and Wang et al. [16], we comprehensively explore the space of various analysis parameters and decisions, including a wider variety of data sets, choice of the differential expression analysis tool, fold change filtering method, and impact on downstream gene set enrichment analysis. Moreover, we provide a practical bootstrapping procedure to estimate the expected level of replicability and precision from a given data set. This extensive experimentation allows us to formulate recommendations for researchers working with RNA-Seq data sets limited by cohort size.

## Materials and methods

Our main strategy to investigate the replicability of RNA-Seq analysis results is based on repeatedly subsampling small cohorts from large data sets and determining the level of agreement between the analysis results from these subsamples (Fig 1). We obtained such large data sets by querying two public data repositories: The Cancer Genome Atlas (TCGA) and Gene Expression Omnibus (GEO). In total, we obtained 18 data sets, as listed in Table 1.

For each data set and target cohort size $N = \{3, 4, 5, 6, 7, 8, 9, 10, 12, 15\}$, we subsampled 100 RNA-Seq experiments by randomly selecting a small cohort with $N$ replicates from the full data set. These subsampled experiments can be interpreted as independent studies aiming to answer the same research question using the same methods but based on different cohorts

**Question 1: How reliable are RNA-Seq analysis results obtained from small cohorts?**

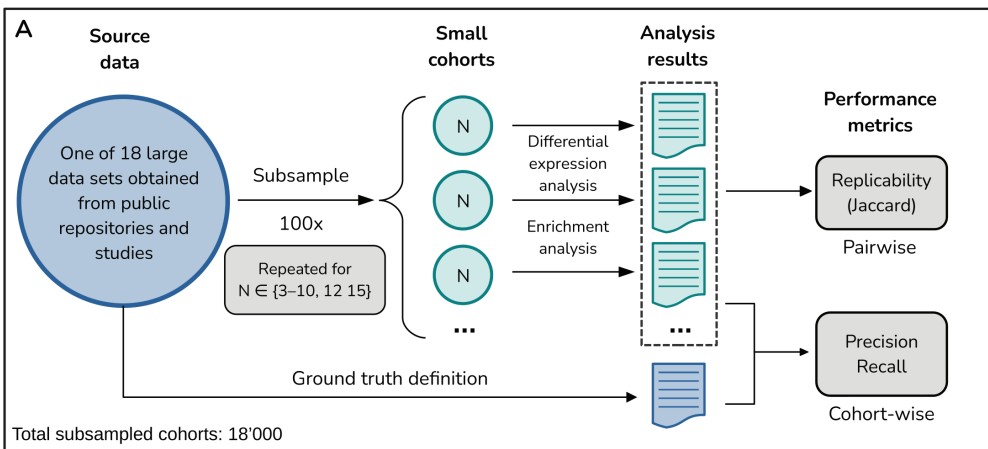

**Question 2: Given a small cohort, can we use bootstrapping to predict the expected level of reliability?**

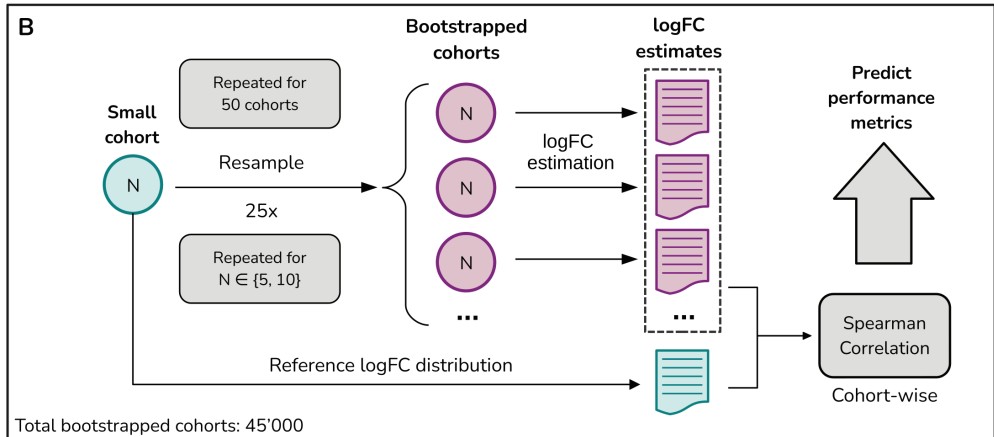

**Fig 1. Flowchart of the study design.** *Top panel:* A large RNA-Seq data set obtained from a public repository is subsampled to yield 100 small cohorts of size *N*, which are analyzed separately and compared pairwise to measure the level of agreement between the results. Additionally, the same analysis steps are run on the entire source data set to define the ground truth, from which precision and recall of results can be computed. This procedure is repeated for 18 different data sets and ten different cohort sizes ranging from 3 to 15 (18'000 cohorts in total). *Bottom panel:* For a given small cohort, the replicates are resampled with replacement to yield 25 bootstrapped cohorts. The log fold change distributions of the bootstrapped cohorts are compared against the original cohort using Spearman's rank correlation, which is used as a predictor in a linear regression model to predict the performance metrics from the top panel.

drawn from the overall population. Although a given sample may appear in multiple subsampled cohorts, each cohort internally consists of unique samples. In total, we subsampled 18'000 cohorts and analyzed each one using multiple analysis pipelines. For data sets with paired samples (i.e. normal and tumor tissue samples from the same donor), our subsampling preserved the pairing of the samples.

## RNA-Seq data sets

Table 1 lists the data sets used in this study. All data sets were downloaded from public repositories as matrices of pre-processed, unnormalized integer read counts.

**Table 1. Summary of the used RNA-Seq data sets.**

| Data | Scenario | Description | Design | Replicates |
|------|----------|-------------|--------|------------|
| BRCA | Normal–Tumor | Breast invasive carcinoma | Paired | 111 |
| KIRC | Normal–Tumor | Kidney renal clear cell carcinoma | Paired | 72 |
| THCA | Normal–Tumor | Thyroid carcinoma | Paired | 58 |
| LUAD | Normal–Tumor | Lung adenocarcinoma | Paired | 52 |
| LUSC | Normal–Tumor | Lung squamous cell carcinoma | Paired | 51 |
| PRAD | Normal–Tumor | Prostate adenocarcinoma | Paired | 51 |
| LIHC | Normal–Tumor | Liver hepatocellular carcinoma | Paired | 50 |
| COAD | Normal–Tumor | Colon adenocarcinoma | Paired | 39 |
| LMAB | Tumor–Tumor | Breast luminal A vs. luminal B | Controlled | 161 |
| BSLA | Tumor–Tumor | Breast basal vs. luminal A | Controlled | 126 |
| BSLB | Tumor–Tumor | Breast basal vs. luminal B | Controlled | 126 |
| BSHR | Tumor–Tumor | Breast basal vs. HER2+ | Controlled | 59 |
| HRLA | Tumor–Tumor | Breast HER2+ vs. luminal A | Controlled | 59 |
| HRLB | Tumor–Tumor | Breast HER2+ vs. luminal B | Controlled | 59 |
| GIPF | Normal–Disease | Idiopathic pulmonary fibrosis | Controlled | 102 |
| GATB | Normal–Disease | Active tuberculosis | Not controlled | 50 |
| HSPL | 1st–3rd trimester | Human placenta | Controlled | 43 |
| SNF2 | Wild type–Mutant | Yeast SNF2 mutation | Not controlled | 42 |

The data sets used in this study are grouped into three scenarios: normal vs. tumor tissue samples from TCGA, tumor vs. tumor tissue samples from TCGA, and miscellaneous non-cancer data sets. The replicate column lists the number of samples in each condition (control vs. perturbed). The design column lists the experimental design used to control for confounders. Paired-design data have matching normal and tumor tissue samples from the same patient. Controlled data are unmatched but use clinical variables as covariates. Two data sets are not controlled for confounders, in accordance with the original studies from which we obtained them.

We downloaded the TCGA data sets using a custom Python notebook that accessed the API of the Genomic Data Commons [23] of the United States National Cancer Institute. For each primary cancer site, we filtered the available cases by experimental strategy (RNA-Seq) and data category (transcriptome profiling). For improved statistical power [10,24], we focused on paired design experiments with one normal tissue sample and one matching primary tumor sample for each patient. There were eight projects with at least 50 patients. To avoid excessive cohort heterogeneity, we kept only patients with the most common disease type for the given project.

In addition to normal vs. matched tumor comparisons, we downloaded unmatched breast cancer (BRCA) tissue samples to perform comparisons between tumor tissues. To this end, we grouped the BRCA samples into four subtypes: luminal A, luminal B, basal-like, and HER2-enriched. The subtype labels were obtained from a prior study that used the same TCGA data [25]. To reduce the number of confounding factors (crucial when cohort sizes are low), we removed a minority of samples originating from male donors.

We also queried GEO for non-cancer data with sufficiently many samples and identified three datasets with unmatched samples. The GATB data set (series accession number GSE107994 [26]) compares control samples vs. patients with active tuberculosis. The GIPF data set (GSE150910 [27]) compares control samples vs. patients with idiopathic pulmonary fibrosis. The HSPL data set (series accession number GSE247382 [28]) compares first vs. third trimester human placenta samples. Finally, we included a *Saccharomyces cerevisiae* (yeast) data set that was used in a prior methodological study on differential expression analysis [7]. This data set compares wild type samples vs. samples with mutated SNF2 gene, which is part of a transcriptional activator that brings about significant changes in transcription.

The median number of replicates across all 18 data sets is 58.5 (range 39–161). For each data set, we filtered lowly expressed (and hence uninformative) genes using the `filterByExpr` function from edegR [4].

### Differential expression analysis

For each subsampled experiment, we determined the genes that are differentially expressed between control and perturbed samples using the popular R packages edgeR [4] and DESeq2 [3]. Both packages rank among the leading tools when considering small sample sizes [7] and overall performance [10], boasting 39'962 and 76'979 respective citations on Google Scholar as of February 20, 2025.

Before testing for differential expression, we normalized the counts using the `calcNormFactors` function from edgeR and `estimateSizeFactors` from DESeq2, respectively. For the normal-tumor scenario, a paired-sample design matrix was used to improve statistical power and control for patient-level confounders. For the tumor-tumor scenario, the samples were unmatched and we instead controlled for age and tumor purity. For the remaining data sets, we used the respective original studies to determine the covariates to control for. Specifically, for the GIPF data set, we controlled for age, sex, and smoking history. (Race was removed as a fourth covariate as our study investigates small cohort sizes. For the smallest cohort sizes $N = \{3, 4, 5\}$, having four covariates yields rank-deficient design matrices in most cases, making the analysis impossible. A random forest feature importance analysis using scikit-learn [29] revealed race to be the least informative of the four covariates.) For the HSPL data set, we controlled for fetal sex. For the GATB and SNF2 data sets, no additional covariates were considered.

Unless otherwise noted, we determined the significant DEGs using a 5% threshold on the Benjamini-Hochberg adjusted $p$-values to control the false discovery rate (FDR); we will henceforth use the term "FDR" to refer to the adjusted $p$-values. To test for differential expression, we considered several statistical approaches, as listed in Table A in S1 Text and described in the next two paragraphs.

First, an important issue that needs to be addressed is the question of a minimum absolute fold change, below which genes are not considered to be of interest. There are two approaches to filtering genes with small fold change. The statistically principled way is to formally test the null hypothesis $|\log_2 \text{FC}| \leq t_{null}$, where $t_{null}$ is the chosen minimum significant threshold [30]. However, many practitioners instead use an alternative approach that we will designate as post hoc thresholding (also called double filtering in [31]). In this approach, only the null hypothesis of a zero fold change is formally tested, followed by a filtering step in which the significant genes whose estimated fold change is below a chosen cutoff are removed. One disadvantage of this approach is that the FDR is no longer properly controlled at the specified significance level. Compared to formal statistical testing, post hoc thresholding is a more permissive approach, as genes that barely pass the threshold might not reach statistical significance in a formal test. For concision, we focus on formal thresholding in the main text and include results with post hoc thresholding in S2 Text.

For DESeq2, we used the default Wald test, which tests for differential expression above a user-specified absolute $\log_2$ fold change threshold. For edgeR, there are a variety of statistical tests available. When not using a fold change threshold, two of the available methods include the likelihood-ratio test (LRT) and the quasi-likelihood F-test (QLF). The QLF test is described by the authors as offering more conservative and reliable type I error control when the number of replicates is small [32]. However, when using a formal fold change threshold, the authors of edgeR recommend the t-test relative to a threshold (TREAT) [30]. Like the

other methods, TREAT is a parametric method that requires negative binomial models to be fitted to the data before any testing can be done. Users can choose between the functions `glmFit` or `glmQLFit`, which are the same functions used for the LRT and QLF pipelines, respectively. The TREAT implementation in edgeR detects which function was used and conducts a modified LRT or QLF test accordingly. For this reason, we use the labels LRT and QLF to designate our use of TREAT.

To summarize, we used three statistical tests: DEseq2 Wald, edgeR LRT, and edgeR QLF. For each test, we used a formal fold change threshold of $t_{null} = 1$ (results shown in main text), as well as $t_{null} = 0$ with and without a post hoc threshold of $t_{post} = 1$ (results shown in S2 Text).

## Enrichment analysis

Subsequently to running differential expression analysis for each subsampled cohort, we performed Gene Set Enrichment Analysis (GSEA) [6]. Specifically, we used the Python package GSEApy [33] in preranked mode. This method takes as input a list of all genes expressed in the experiment, ranked by a user-provided metric, such as log fold changes (logFC) or signed log $p$-values. The method then tests whether a given biologically annotated gene set is enriched at the extreme ends of the ranked gene list, taking into account the magnitudes of the provided metric.

To save computing resources, we limited our investigation of GSEA results to cohort sizes $N \in \{3, 5, 7, 10, 12, 15\}$. For the ranking metric, we used the logFC, which is a popular choice and corresponds to the default parameter of GSEApy in standard (not preranked) mode. However, for ranking genes by logFC, the DESeq2 authors recommend using one of their included shrinkage estimators to obtain more stable logFC estimates. Therefore, we computed shrunken logFC estimates using the adaptive shrinkage (ashr) [34] option in DESeq2. It is important to keep in mind that shrunken logFC estimates are only used for ranking genes and not for calling DEGs, in accordance with recommendations by the DESeq2 authors.

For the gene sets, we used libraries curated by Enrichr [35] and accessed from GSEApy. Specifically, we tested for enriched pathways from the Kyoto Encyclopedia of Genes and Genomes (KEGG), as well as enriched Gene Ontology (GO) terms from the Biological Process subdomain (BP). For human data, we used `KEGG_2021_Human` and `GO_Biological_Process_2023` libraries. For yeast data, we used `KEGG_2019` and `GO_Biological_Process_2018` from the corresponding YeastEnrichr database. To determine the significance of the gene sets, we again used a 5% threshold on the Benjamini-Hochberg adjusted $p$-values to control the FDR.

## Experiment replicability

We proceed by introducing a fundamental performance metric used throughout this paper. The goal is to measure the level of mutual agreement between the results obtained from subsampled experiments with the same cohort size (Fig 1). For this, we designate the set of analysis results as $S_i$, where $S$ can be a set of DEGs, a set of enriched GO terms, or a set of enriched KEGG pathways, and $i \in \{1, 2, \ldots, 100\}$ indexes the experiment. We then use the Jaccard index (intersection over union) to define the inter-experiment replicability of results from two subsampled experiments,

$$\text{Replicability}(i, j) = \frac{|S_i \cap S_j|}{|S_i \cup S_j|}. \tag{1}$$

This value is between zero (no overlap of results) and one (perfect agreement). If either $S$ is empty, we define the replicability to be 0. We report the median experiment replicability over all $\binom{100}{2}$ = 4950 distinct pairs $(i,j)$.

## Ground truth and binary classification metrics

An analysis pipeline consists of a fold change thresholding method (zero, formal, post hoc) and a statistical test for DEG identification (edgeR QLF, edgeR LRT, DESeq2 Wald). In addition to measuring the experiment replicability, we defined pipeline-specific ground truths by running the respective pipeline on the full data set with all replicates (Table 1). This yields a list of ground truth DEGs and their corresponding ground truth logFC estimates. Similarly, the ground truth of enriched gene sets was obtained by running the enrichment analysis using shrunken logFC estimates from the full data set.

After having determined the ground truth for both DEGs and enriched gene sets, we computed two classical binary classification metrics: precision (positive predictive value) and recall (sensitivity). This approach enables us to quantify the extent to which results from small cohort sizes approximate those obtained from much larger cohorts. The metrics are defined as

$$\text{Precision} = \frac{\text{TP}}{\text{TP} + \text{FP}},$$
$$\text{Recall} = \frac{\text{TP}}{\text{TP} + \text{FN}},$$

where TP = true positive, FP = false positive, TN = true negative, FN = false negative. The precision is undefined when the denominator is zero (i.e., when no genes pass the significance threshold) and excluded from calculations of summary statistics like the median.

## Bootstrapping small cohorts

Our study design is based on creating small cohorts by subsampling large RNA-Seq cohorts. To address the needs of a practitioner analyzing a single small cohort, we propose a simple resampling strategy to estimate the expected reliability of differential expression and enrichment results. The procedure is based on the well-established statistical technique of bootstrapping [36] and illustrated in Fig 1B. Given a data set with $N$ distinct replicates, a bootstrapped count matrix is created by resampling with replacement $N$ "new" replicates from the original set of $N$ replicates. For paired-design data sets, matched samples are always resampled jointly; for unpaired data sets, each condition is resampled separately. Next, the bootstrapped data set is used to compute logFC estimates with DESeq2. The bootstrapped logFC estimates are then compared with the estimates from the original data set to assess their variability. Specifically, each gene is ranked according to its logFC, upon which the Spearman rank correlation is calculated between the bootstrapped and original rankings. A low correlation indicates that the fold change estimates are sensitive to perturbations in the cohort composition. As we are interested in measuring the intrinsic level of variability of the given data set, we do not use a shrinkage estimator for the logFC, as it would reduce the variability by design.

The entire bootstrap procedure is repeated for $k$ trials to estimate the mean Spearman correlation. In our case, we chose $k$ = 25 trials to limit the computational load. However, in real-world scenarios, practitioners typically only have a small number of data sets, so the number of trials can be readily increased. To save computing resources, we demonstrate the bootstrapping using DESeq2 only. Finally, we restricted our assessment to 50 cohorts of size $N$ = 5 and $N$ = 10 per data set, for a total of 45'000 bootstrapped cohorts.

## Results

### Comparison of statistical tests

The empirical ground truth of DEGs for a given statistical test (QLF, LRT, Wald) and fold change threshold is derived by analyzing all replicates in a given data set. Fig A in S2 Text shows the number of DEGs for different tests and fold change thresholds. Depending on the data set and thresholding method, the ground truth varies over two orders of magnitude, from 120 to 14'730 DEGs. The number of ground truth DEGs remains relatively consistent across different statistical tests, with a mean Jaccard index of 0.87, indicating strong agreement. Fig A in S1 Text compares the three tests for results derived from subsampled cohorts. LRT and Wald tests perform comparably; however, the QLF test is often too conservative to be used for the smallest cohort sizes, although it offers the highest precision. All three tests perform comparably for the SNF2 data set. Regarding fold change thresholding strategies, the formal approach offers more reliable type I error control than post hoc filtering. Moreover, not using any thresholds at all yields an impractically large number of DEGs when cohort sizes are large. For the remainder of this text, we will thus focus on results derived using the Wald test with a formal threshold of $|\log FC| > 1$. Results for other tests and thresholds are shown in Fig B–I in S2 Text.

### DEG performance metrics

We proceed by exploring how our performance metrics for DEGs vary with the cohort size $N$. Fig 2A shows the median replicability of 100 subsampled cohorts as a function of the cohort size. Except for the SNF2 data set, all data sets show low (<0.5) replicability for the smallest cohort size of $N = 3$. For the largest cohort size of $N = 15$, we observe a wide range of replicability values depending on the data set. The median number of DEGs is shown in Fig 2B.

Comparing the precision and recall of DEGs (Fig 2C and 2D), we observe that the precision rises more steeply than recall for small cohort sizes. Specifically, we observe that 10 out of 18 data sets (SNF2, GATB, HSPL, and all normal-tumor data sets except PRAD) exceed the precision of 0.9 for $N > 5$, of which 7 data sets (except GATB, LIHC, and LUAD) reach the target precision of $1 - 5\%$ FDR = 0.95. In contrast, for all data sets except SNF2, recall is below 0.5 for $N < 7$. From these observations, we conclude that false negatives (low recall) are a more significant driver of low replicability than false positives (low precision).

Among the 18 data sets, we identify two data sets that represent the extreme ends of observed performance metrics: SNF2 (best performing) and LMAB (worst performing). We characterize these two data sets in the next section.

### Population heterogeneity and fold change inflation

Fig 3A and C show heat maps of sample correlations for the SNF2 and LMBA data sets. Correlations were computed using the logCPM (counts-per-million) values estimated from count matrices normalized using DESeq2. Heat map rows and columns were ordered using hierarchical clustering with Ward's method in SciPy [37]. We observe from panel A that the SNF2 samples cluster perfectly into the two conditions (wild type and mutant), with high population homogeneity within each condition. By contrast, the LMAB samples cluster very poorly into the two conditions (luminal A and luminal B), with high heterogeneity among the samples. These findings are consistent with expectations, given that the SNF2 samples originate from cell colonies, whereas the LMAB samples are derived from heterogeneous tumor tissues. Moreover, the luminal A vs. luminal B samples are relatively more similar than the other data

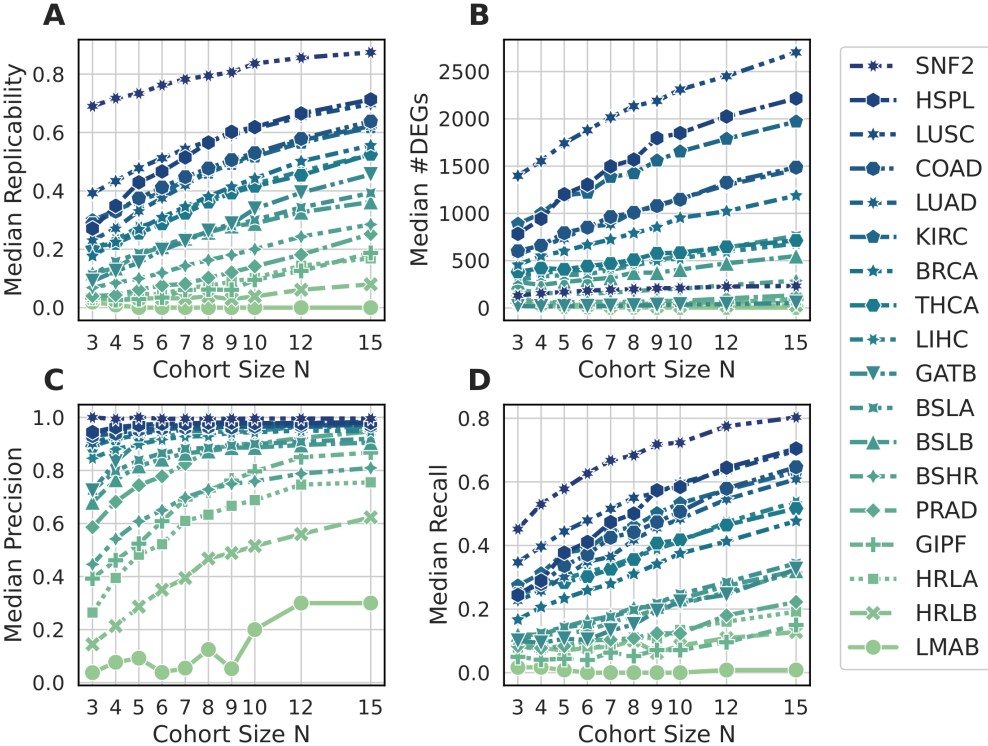

**Fig 2. DEG performance metrics as a function of the cohort size.** Each symbol summarizes the median of 100 cohorts. All panels show results using the DESeq2 Wald test with |log$_2$ FC| > 1. Results using other tests and fold change thresholds are shown in S2 Text.

sets with tumor comparisons, resulting in the worst cluster separation of conditions among all data sets (see Fig A–H in S3 Text for additional heat maps).

The blue symbols in Fig 3B and 3D show the ground truth logFC estimates for all genes expressed in the SNF2 and LMAB data sets (disregarding a small fraction of genes for which DESeq2 was unable to compute the logFC). The interquartile range (IQR) of logFC estimates from all 100 subsampled cohorts is also shown as gray bars for $N = 3$ and red bars for $N = 15$. For the SNF2 data set, the estimates from subsampled cohorts show little variability even for the smallest cohort size of $N = 3$. However, for the LMAB data set, the $N = 3$ estimates show substantial variability, with 31.2% of all genes having an IQR that crosses the absolute logFC threshold of 1 (which we use to define DEGs). For $N = 15$, the number of genes that cross the threshold drops to 8.99%, which is still twice as large as the corresponding number for the SNF2 data set with $N = 3$ (4.36%). Although most of the other data sets have a comparable or even higher number of crossings as LMAB (Fig I–P in S3 Text), LMAB has the smallest number of DEGs in the ground truth (Fig A in S2 Text). Thus, for small cohorts, the fraction of spurious findings from inflated logFC estimates is large, resulting in poor precision.

To summarize, it is not surprising that the SNF2 and LMAB exhibit the highest and lowest precision in Fig 2, respectively. The SNF2 data set is so homogeneous and well-separated by condition that the subsampling has little influence on the logFC estimation, which has little variance even for the smallest cohort sizes. By contrast, the LMAB data has few true DEGs and the logFC estimates exhibit substantial sampling variance, which leads to logFC estimates that are either inflated or deflated. In the case of inflation of a non-DEG, the respective

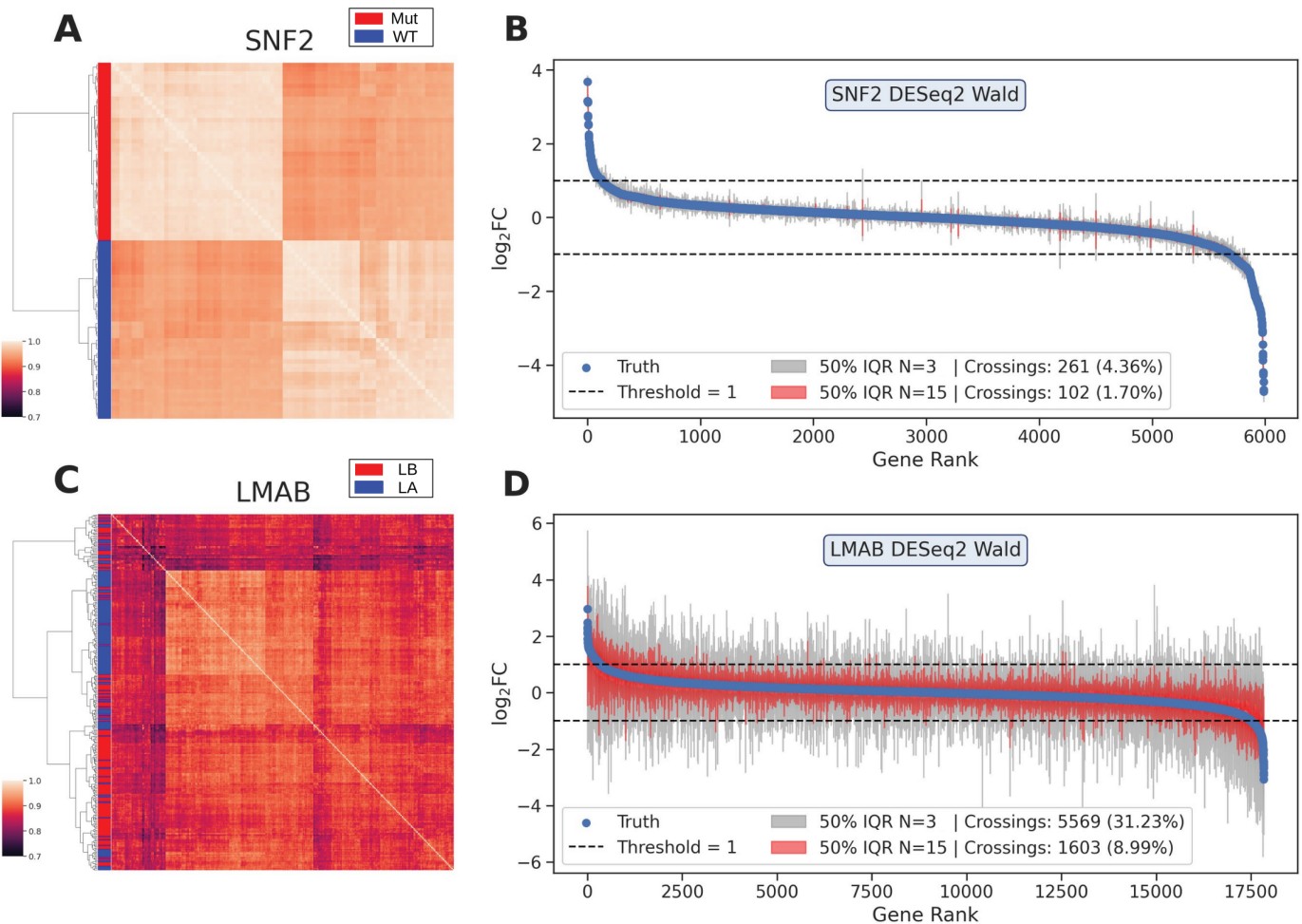

**Fig 3. Heat maps and fold change estimates for SNF2 and LMAB data sets.** *Left column:* Heat maps showing the logCPM correlation of samples for the SNF2 and LMAB data sets. Heat map rows and columns were ordered using hierarchical Ward clustering. *Right column:* Fold change estimates of expressed genes in the SNF2 and LMAB data sets. Blue dots represent the ground truth estimate from the full data set. Gray (red) bars represent the interquartile range of estimates obtained from 100 subsampled cohorts of size $N = 3$ ($N = 15$). The horizontal dashed line shows the logFC threshold used to define DEGs. The legend lists the number of bars that cross the dashed lines.

gene is more likely to spuriously pass significance and fold change thresholds, thus yielding a false positive. This effect is also known as *regression to the mean* in statistics. More generally, the potential of underpowered studies to inflate effect sizes and undermine the reliability of results has been reported in a range of fields [12,38,39].

## Enrichment performance metrics

Fig A in S2 Text shows the number of enriched terms (significant gene sets) in the ground truth for KEGG and GO libraries. Fig 4 in the main text shows the performance metrics for enriched terms from the GO biological process subdomain. Results from KEGG are qualitatively similar and shown in Fig J in S2 Text. All figures show results obtained from logFC estimates with adaptive shrinkage.

Comparing Figs 2C and 4C, we observe that the precision of enriched terms is generally worse than the precision of DEGs (for $N = 15$, median DEG precision is 0.95 and median

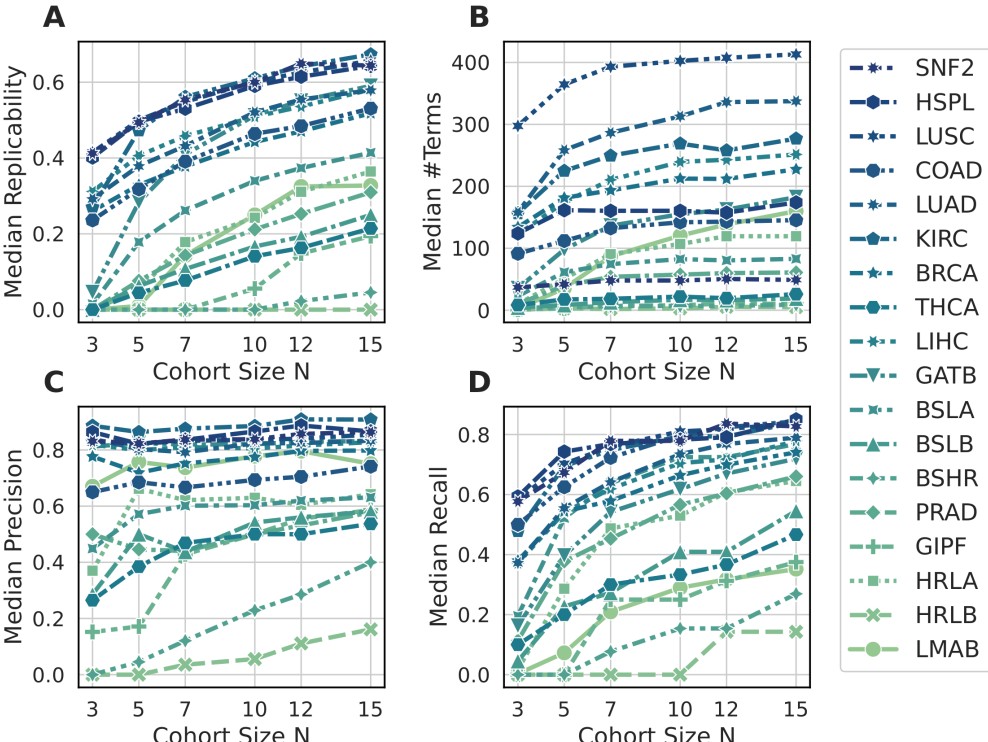

**Fig 4. Enrichment performance metrics as a function of the cohort size.** Each symbol summarizes the median of 100 cohorts. All panels show enriched terms from the GO biological process subdomain.

enrichment precision is 0.75). In contrast, enriched terms exhibit generally better recall (for $N = 15$, median DEG recall is 0.41 and median enrichment recall is 0.73). Median replicability is similar between DEGs and enriched terms.

Notably, the LMAB data set shows much higher precision of enriched terms compared to the precision of DEGs. From Fig A in S2 Text we also observe that the LMAB data set has one of the largest enrichment signals in the ground truth, despite being the data set with the fewest DEGs above the fold change cutoff. This suggests that data sets unsuitable for differential expression analysis are not necessarily unsuitable for enrichment analysis.

Fig U–W in S2 Text show box plots comparing enrichment metrics obtained from shrunken and unshrunken logFC estimates. We observe that shrunken logFC estimates yield moderately higher precision for $N = 3$, at the cost of lower recall and replicability. For $N = 15$, shrinkage has little effect on the metrics.

Overall, we observe a substantial range of possible outcomes depending on the data set, whether we look at DEGs or enriched terms. This makes it particularly problematic to use low-powered cohorts in real-world scenarios unless practitioners can estimate the likely performance regime of their data sets. We will address this question in the next section.

## Bootstrapping

We continue by demonstrating a practical approach to assist researchers with low-powered RNA-Seq data sets in determining the expected reliability of their results. The idea is to repeatedly bootstrap-resample a given data set and compute the Spearman rank correlation coefficient of gene logFC rankings between the bootstrapped data sets and the original data

set (referred to as Spearman correlation below). A low correlation indicates that the logFC estimates are sensitive to changes in the cohort composition, from which we expect lower precision and replicability.

From Fig 5, we observe that the Spearman correlation is indeed a good predictor for the precision, recall, and replicability of the identified DEGs. In particular, a heuristic threshold of Spearman $\gtrsim 0.9$ indicates high precision ($\gtrsim 0.9$). Conversely, results from data sets with Spearman $\lesssim 0.8$ should be interpreted with caution, as their precision can be low and the identified DEGs likely cease being significant when more samples are added to the cohort.

Fig K–L in S2 Text show the same figure for enriched gene sets from KEGG and GO databases, respectively. The results are qualitatively similar to those obtained for DEGs, albeit with moderately lower predictive power for precision. Nonetheless, the associations remain strong enough that it would be prudent to run the bootstrapping procedure before performing enrichment analysis with low-powered data sets.

Fig M–R in S2 Text show equivalent figures using two non-bootstrapped statistics that can be calculated directly from the original cohort: the number of DEGs and the standard deviation of the logFC distribution. Both of these metrics measure the signal strength present in the data, and one would expect lower precision and recall from weaker signals. However, in all tested scenarios, these two statistics emerge as substantially worse predictors than the bootstrapped Spearman correlation. The relative performance of the three different statistics is also summarized in Fig S in S2 Text.

The symbols in Fig 5 show the median Spearman correlation over 50 cohorts. However, if the correlation varies too much between cohorts, it ceases to be a useful predictor for any given individual cohort. Therefore, we show in Fig TA and TC in S2 Text the variability of the Spearman correlation over 50 cohorts for each data set, for $N \in \{5, 10\}$, as well as the corresponding precision. The figure shows that data sets with low precision rarely produce cohorts that have a high Spearman correlation by chance. Conversely, data sets with high precision rarely produce cohorts with low Spearman correlations. Fig TB and TD in S2 Text show scatter plots of the precision and Spearman correlations for all data sets combined. From these data points, we can calculate the empirical probability of the precision exceeding a given threshold, conditioned on the Spearman correlation exceeding a given threshold. For example, for $N = 10$, a Spearman correlation >0.9 results in precision >0.9 in 96% of cases, and precision <0.8 in 1% of cases. More examples are given in Section 1.6 in S2 Text.

## Discussion

We have comprehensively analyzed the replicability and reliability of RNA-Seq differential expression results obtained from small cohorts. Compared to recent work by Cui et al. [14], we used a wider variety of data sets, considered multiple tools for the analysis, characterized fold change thresholding strategies, and evaluated the impact of low replicability on downstream enrichment analysis (see Table B in S1 Text for a detailed comparison of the methodology used). We support their conclusion that differential expression results obtained from small cohorts ($N \lesssim 10$) generally lead to low inter-experiment replicability (they use the term "overlap rate"). In contrast to [14], our results show that low replicability does not necessarily imply that DEGs do not generalize to larger cohorts. Depending on the level of sample heterogeneity in the overall population, data sets with a small number of replicates can still achieve high precision, despite low recall and replicability. We further show that our proposed bootstrapping procedure can successfully predict the precision, recall, and replicability of DEGs and enriched gene sets for our tested data sets.

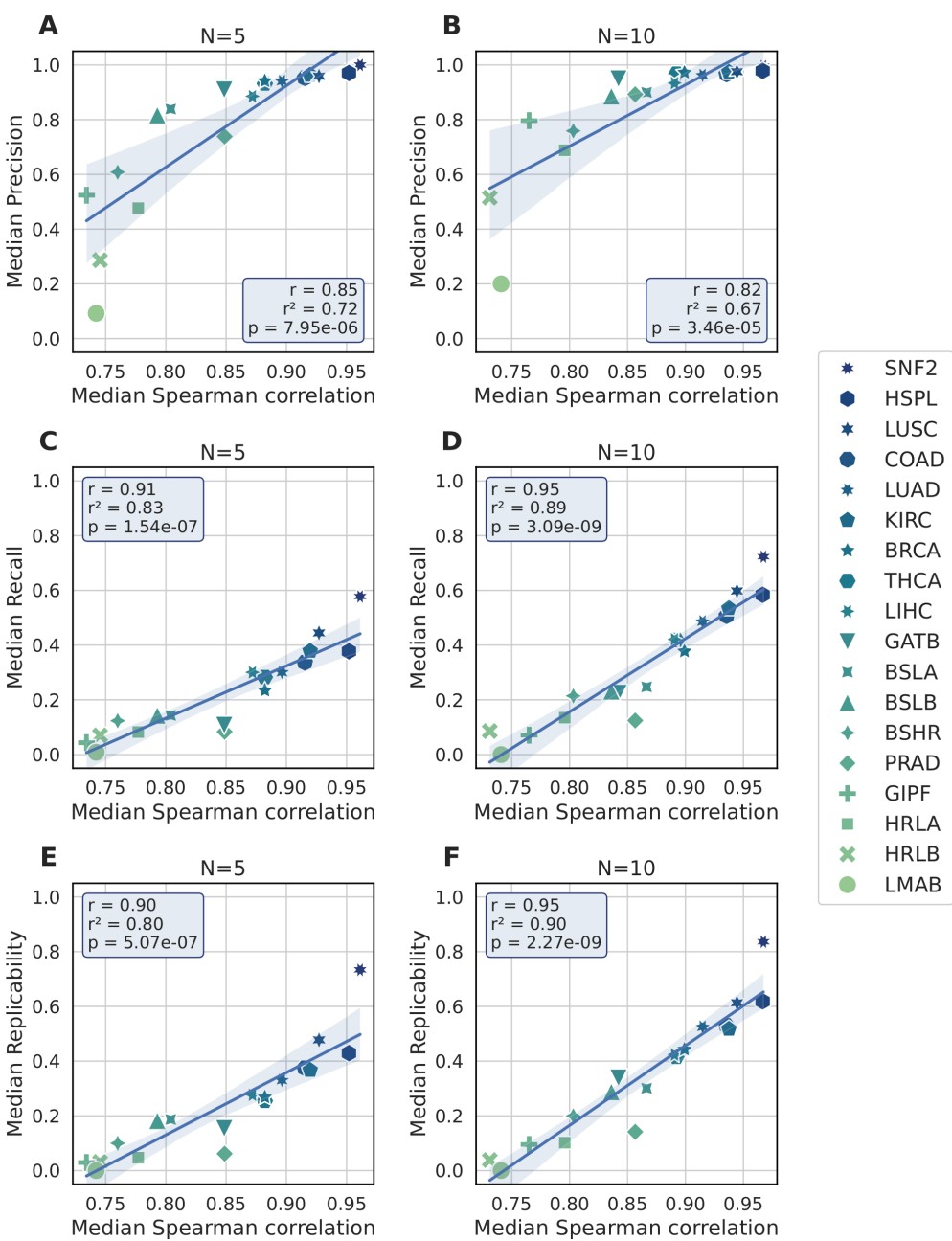

**Fig 5. Bootstrapping results.** Performance metrics (precision, recall, and replicability) of DEGs versus bootstrapped Spearman logFC correlation for cohort sizes $N \in \{5, 10\}$. Performance metrics are as in Fig 2. Each symbol summarizes the median of 100 cohorts on the $y$-axis and 50 cohorts on the $x$-axis. The boxes list the Pearson correlation coefficient $r$, the coefficient of determination $r^2$, and the $p$-value from testing the null hypothesis that the distributions underlying the data points are uncorrelated.

Li et al. [40] recently reported that edgeR and DESeq2 suffer from false positives, as demonstrated by the high number of DEGs they identify in permuted input data sets where significant differences are assumed to be removed by permutations. This issue does not seem to be relevant for the RNA-Seq data analyzed here as: (1) DEGs start to appear in

permuted data only when the number of samples is large ($n \gtrsim 32$) and (2) the numbers of DEGs thus found are much smaller compared to the numbers of DEGs found in the unpermuted data (Fig G in S1 Text). In [40], the most striking observations have been made for an immunotherapy study. We have noticed that some highly expressed genes in this data set have several zero counts that are known to cause problems for edgeR [41]; this could have contributed to the reported behavior.

We proceed with a note on the effect of our subsampling procedure. First, we note that our results are based on relatively small parent data sets (median $N = 58.5$). This means that our subsampled cohorts inevitably contain shared replicates between cohorts. The most extreme case is given by the largest subsampled cohort size of $N = 15$ for the smallest data set, COAD, with 39 replicates. In this case, there are $\binom{39}{15} \approx 2.5 \cdot 10^{10} \gg 100$ possibilities for subsampling distinct cohorts. However, the expected number of replicates shared between two subsampled cohorts is $15^2/39 \approx 6$. These shared replicates inflate our computed replicability metrics by a small amount, compared to a scenario where no two replicates have identical expression counts. However, Fig B in S1 Text shows that this inflation is minor, on the order of $0.1 - 1\%$.

A similar consideration can be made for our ground truth definition. Since the samples in a given subsampled cohort also contribute to the ground truth definition from which we calculate our performance metrics, there is a degree of circularity in the analysis. However, when the parent data sets are sufficiently larger than the subsampled data sets, this effect is negligible (see Fig C in S1 Text). An alternative strategy that entirely avoids circularity would be to exclude the subsampled cohort from the ground truth definition, yielding separate ground truths for each subsampled cohort (similar to the technique of cross-validation in statistics). However, such a study design would require substantially more computing resources, which is likely not worth the effort given the minimal expected gain.

The main limitation of our study is its focus on human tissue samples; the SNF2 data set (yeast cell culture) is the only exception. The generalizability of our results to other sample types and organisms remains to be tested. However, because of the broad applicability of RNA-Seq experiments, it is impossible to fully answer this question in the scope of a single study. Moreover, our study is limited to bulk RNA-Seq data. Generalizability to newer technologies, such as single-cell and spatial RNA-Seq, remains to be tested. However, due to the substantial increase in complexity and computational requirements of single-cell and spatial analysis pipelines, an analogous replicability study based on repeatedly subsampling cohorts would be challenging. Instead, we highlight a recent relevant study by Squair et al. [42], who showed that statistical methods for differential expression analysis that do not account for biological variability are prone to false discoveries in single-cell data.

As our study design is computationally demanding, we limited the exploration of enrichment analysis results to the simple but popular case of GSEA with genes preranked by logFC. However, enrichment analysis is a complex topic with many researcher degrees of freedom, including many choices for the gene ranking metric. Alternative metrics include $\text{sign}(\log_2 \text{FC}) \times \log_{10}(p\text{–value})$ or computing the signal-to-noise ratio from the count matrix [43]. In place of a subsample-based study design, we are currently conducting a study to enhance the robustness of enrichment analysis results using an ensemble learning approach that integrates multiple tools and ranking metrics.

## Conclusion

Although the broader replication crisis in science [20–22,44,45] includes numerous human factors such as dysfunctional incentive systems, selective reporting, inadequate statistical training, and publication bias, here, we assumed otherwise ideal research practices and only

concerned ourselves with low replicability arising from underpowered studies of heterogeneous biological populations. Our findings suggest that most RNA-Seq differential expression results obtained from small cohort sizes ($N \lesssim 10$) are unlikely to be confirmed in replication experiments. However, we also observe that low replicability of DEGs does not necessarily imply a high prevalence of false positives, as false negatives are a more significant driver of low replicability. For enrichment results, we find lower precision and higher recall compared to DEGs. In general, there is substantial variability in performance metrics depending on the characteristics of the data set, with some data sets achieving high precision even for relatively small cohort sizes. Therefore, practitioners of RNA-Seq analysis with low-powered cohort sizes run the risk of erroneous research unless they can estimate the likely performance regime of their data sets. To this end, we successfully used a simple bootstrapping procedure to estimate from a given small cohort whether the results are likely to have an inflated number of false positives, and what level of replicability to expect. We conclude with a summary of recommendations for practitioners working with RNA-Seq data obtained from small cohorts:

- Significant DEGs from one small cohort are unlikely to be significant in another small cohort (low replicability), unless it is known that the population is very homogeneous (e.g. cell cultures).
- Calculating Spearman correlations using the bootstrap procedure described in this study may help with assessing what level of replicability and precision to expect. If the observed Spearman correlation is >0.9, the data set is robust to perturbations in the cohort composition, likely resulting in high precision and comparatively higher replicability. If the correlation is <0.8, the data set is sensitive to perturbations, likely resulting in low precision and replicability; thus, results should be interpreted with caution. A Python workflow to perform the bootstrapping and Spearman calculation is available via GitHub (https://github.com/pdegen/BootstrapSeq).
- If only DEGs above a minimum fold change are of interest, we recommend statistically testing for differential expression exceeding this threshold for better type I error control, rather than post hoc filtering of DEGs.
- The DESeq2 Wald and edgeR LRT tests with formal fold change thresholds perform comparably in our evaluation. Unless the data is very homogeneous or contains a strong signal, the edegR QLF test is typically not powerful enough for very small cohorts $N \lesssim 5$, but offers the highest precision in case any DEGs are detected. Thus, for confirmatory analyses, we recommend QLF, whereas for exploratory analyses, we recommend either Wald or LRT.

## Supporting information

**S1 Text. Additional tables and figures.** Table A: Statistical tests used for differential expression analysis. Table B: Comparison of Cui et al. [14] with this study. Fig A: Performance metrics for different statistical tests. Fig B: Influence of subsampling with replacement on replicability. Fig C: Influence of subsample inclusion in ground truth on precision and recall. Figs D–F: Partial results with Wilcoxon signed-rank test. Fig G: DEGs from 8 permuted and unpermuted data sets.
(PDF)

**S2 Text. Additional figures.** Fig A: Ground truth size. Figs B–I: DEG performance metrics for additional tests and fold change thresholds. Fig J: KEGG enrichment performance metrics. Figs K–L: Bootstrapping results for enrichment analysis. Figs M–R: Predicting performance metrics from non-bootstrapped statistics. Fig S: Comparison of predictor statistics. Fig T: Variability of Spearman correlations. Figs U–W: Enrichment metrics for shrunken vs. unshrunken logFC.
(PDF)

**S3 Text. Additional figures.** Fig A–H: Heat maps for the remaining data sets (not including SNF2 and LMAB). Fig I–P: Fold change figures for the remaining data sets.
(PDF)

## Acknowledgments

Calculations were performed on UBELIX (http://www.id.unibe.ch/hpc), the HPC cluster at the University of Bern. The results published here are in part based upon data generated by the TCGA Research Network: https://www.cancer.gov/tcga.

## Author contributions

**Conceptualization:** Matúš Medo.

**Data curation:** Peter Methys Degen.

**Formal analysis:** Peter Methys Degen.

**Funding acquisition:** Matúš Medo.

**Investigation:** Peter Methys Degen.

**Methodology:** Peter Methys Degen, Matúš Medo.

**Project administration:** Matúš Medo.

**Software:** Peter Methys Degen.

**Supervision:** Matúš Medo.

**Validation:** Matúš Medo.

**Visualization:** Peter Methys Degen.

**Writing – original draft:** Peter Methys Degen.

**Writing – review & editing:** Matúš Medo.

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
