## [Decision Letter · Decision Letter 0]

28 Jan 2024

Dear Mr. Degen,

Thank you very much for submitting your manuscript "Replicability of bulk RNA-Seq differential expression and enrichment analysis results in cancer research" for consideration at PLOS Computational Biology.

As with all papers reviewed by the journal, your manuscript was reviewed by members of the editorial board and by several independent reviewers. In light of the reviews (below this email), we would like to invite the resubmission of a significantly-revised version that takes into account the reviewers' comments.

The reviewers raised concerns about the simulation methods used in the work, the generalizability of the conclusion from TCGA data only, and some issues in result interpretation. It is suggested that these concerns be fully addressed in a substantially revised manuscript.

We cannot make any decision about publication until we have seen the revised manuscript and your response to the reviewers' comments. Your revised manuscript is also likely to be sent to reviewers for further evaluation.

Sincerely,

Chongzhi Zang

Guest Editor

PLOS Computational Biology

Jian Ma

Section Editor

PLOS Computational Biology

The reviewers raised concerns about the simulation methods used in the work, the generalizability of the conclusion from TCGA data only, and some issues in result interpretation. It is suggested that these concerns be addressed in a substantially revised manuscript.

Reviewer's Responses to Questions

**Comments to the Authors:**

Reviewer #1: Please see the attachment.

Reviewer #2: Degen et al presented a statistical evaluation of the robustness of RNA-seq differential expression analysis using different numbers of samples. The manuscript is well presented. The topic is appreciated by the reviewer. The analysis and comparisons are systematic and well structured. However, a fundamental problem and a minor concern must be solved before this manuscript can be considered for publication.

Main concern: The analysis was purely made on TCGA data. It is unclear if the results can generally capture the robustness level for RNA-seq DEG analysis. The authors are expected to make a systematic analysis on at least five independent RNA-seq data cohorts. It will be better to include samples of different organ types such as blood and brain samples. Also, DESeq and EdgeR are used for DEG for pseudobulk based analysis of scRNA-seq data. It will be good if an evaluation of this part could be included.

Minor concern: Nonparametric Mann Whitney test should be compared.

Reviewer #3: The work performs a replicable analysis for bulk RNA-seq for cancer studies using both simulation studies and real data anlaysis to evaluate how the combination of population heterogeneity and underpowered cohort sizes affects the replicability of RNA-Seq research. However, some conclusions are well-known common sense and other conclusion is not well supported by the study design. Overall, the scientific impact of the study in the RNA-seq design is moderate. I have some comments as follows,

1. The simulation study is random sampling a subset of RNA-seq samples from the total cohorts. However, random sampling may not a good strategies. Covariates of the cohort such as gender, sex may confound the analysis results.

2. In addition the simulation study is not real simulation conceptually but still real data analysis. Usually simulation are generated from a parametric model with underlying gene expression and FC know in advance to evaluate the DE performance under different sample size

3. The major conclusions large sample size results in better reproducibility and small cohort will have poor reproducibility is well-known common sense.

**Have the authors made all data and (if applicable) computational code underlying the findings in their manuscript fully available?**

Reviewer #1: Yes

Reviewer #2: **No: **I did not see codes were provided

Reviewer #3: Yes

PLOS authors have the option to publish the peer review history of their article (what does this mean?). If published, this will include your full peer review and any attached files.

Reviewer #1: No

Reviewer #2: No

Reviewer #3: No
---

## [Decision Letter · Decision Letter 1]

19 Jul 2024

Dear Mr. Degen,

Thank you very much for submitting your manuscript "Replicability of bulk RNA-Seq differential expression and enrichment analysis results in cancer research" for consideration at PLOS Computational Biology. As with all papers reviewed by the journal, your manuscript was reviewed by members of the editorial board and by several independent reviewers. The reviewers appreciated the attention to an important topic. Based on the reviews, we are likely to accept this manuscript for publication, providing that you modify the manuscript according to the review recommendations.

While 2 reviewers do not have further comments, the comments from Reviewer 4 need to be addressed. We suggest that you fully address their comments in a response letter and make substantive clarifications and explanations in the manuscript as necessary. Also, please follow the PLOS policy to make all code and data available.

Sincerely,

Chongzhi Zang

Academic Editor

PLOS Computational Biology

Jian Ma

Section Editor

PLOS Computational Biology

While 2 reviewers do not have further comments, the comments from Reviewer 4 need to be addressed. We suggest that you fully address their comments in a response letter and make substantive clarifications and explanations in the manuscript as necessary. Also, please follow the PLOS policy to make all code and data available.

Reviewer's Responses to Questions

**Comments to the Authors:**

Reviewer #1: The authors have addressed all the points from my review. Their response is thorough and professional, and their revisions (especially to the discussion) will help readers interpret their findings appropriately. I have no additional comments.

Reviewer #2: The authors has addressed all my concerns

Reviewer #4: The manuscript by Degen and Medo presents an analysis of RNAseq replicability and effect size using subsampled partial TCGA data. Although the authors made some changes based on the last round of reviewer comments, some questions are still not well answered. I also have concerns about the foundation of this work.

1) Is it true most cancer studies only have a few replicates?

The assumption of this entire study is based on a previously published paper [PMID: 30428853] which claimed, “A survey by Baccarella et al. [9] reports that about 50% of 100 randomly selected RNA-Seq experiments with human samples fall at or below six replicates per condition, with this ratio growing to 90% for non-human samples.” Unfortunately, Baccarella’s paper is flawed. They used only 100 “random” publications to draw this conclusion, selecting low-impact publications and even several papers from the same team. A random check of a few of these papers reveals miscounted sample sizes, ignored technical replicates, and case studies that never performed DE analysis. Even if we assume Baccarella’s conclusion is true, it is based on all human studies, with only a few cancer studies included. Given that population heterogeneity is a common issue in cancer research, most responsible cancer studies do not use fewer than six replicates. High-impact publications, clinical trials, and consortium studies typically include dozens of replicates per condition. In reality, non-human and non-cancer studies are more likely to fall below six replicates per condition. Overall, the assumption of this paper is incorrect. The authors spent significant effort solving a non-existent problem.

2) The comparison between tumor vs. matching normal lacks practical meaning.

The authors defined the empirical ground truth based on this comparison. Since tumor cells and normal tissue are essentially different cell types, it is common to see a high number of DEGs (“ground truth”), which is rare in other non-human and non-cancer studies. Previous reviewer 1 mentioned this concern, and the authors merely explained it in their rebuttal letter without adequately addressing it. One possibility is that instead of comparing tumor vs. matching normal, they could compare one subtype vs. another subtype, or poor survival vs. better survival. Comparing two groups of tumor samples will not lead to a huge “ground truth” and high precision. The latter comparison is more relevant in clinical or biological research.

3) Some confusion in the description of the data.

“If the cases came from multiple projects, we kept the patients from the most populated project.” As far as I know, no single patient is enrolled in more than one TCGA study. “To avoid excessive cohort heterogeneity, we finally kept only patients with the most common disease type for the given project.” Authors need to provide details about which particular subtype and patient IDs were used in this study. Additionally, they should upload the code to GitHub, along with subsampled patient IDs, to enable reproducibility.

4) Authors ignored other confounding factors previously mentioned by reviewers.

TCGA studies is too complicated. They should include simulated data, non-human data, and non-TCGA data to better understand the utility of this study.

5) This paper is too similar to Cui et al.’s paper [PMID: 35876281], with only a few more TCGA datasets.

If this paper is merely an extension of a previous publication with slightly more assessments and data leading to very similar conclusions, it lacks novelty as a new publication.

**Have the authors made all data and (if applicable) computational code underlying the findings in their manuscript fully available?**

Reviewer #1: Yes

Reviewer #2: Yes

Reviewer #4: **No: **They didn't upload code or data

PLOS authors have the option to publish the peer review history of their article (what does this mean?). If published, this will include your full peer review and any attached files.

Reviewer #1: **Yes: **Kris Sankaran

Reviewer #2: No

Reviewer #4: No

Figure Files:

Data Requirements:

Reproducibility:

References:

---

## [Decision Letter · Decision Letter 2]

24 Oct 2024

PCOMPBIOL-D-23-01720R2Replicability of bulk RNA-Seq differential expression and enrichment analysis resultsPLOS Computational Biology Dear Dr. Degen, Thank you for submitting your manuscript to PLOS Computational Biology. After careful consideration, we feel that it has merit but does not fully meet PLOS Computational Biology's publication criteria as it currently stands. Therefore, we invite you to submit a revised version of the manuscript that addresses the points raised during the review process. Please submit your revised manuscript within 60 days Dec 24 2024 11:59PM. If you will need more time than this to complete your revisions, please reply to this message or contact the journal office at ploscompbiol@plos.org. Please include the following items when submitting your revised manuscript: * A rebuttal letter that responds to each point raised by the editor and reviewer(s). You should upload this letter as a separate file labeled 'Response to Reviewers'. This file does not need to include responses to formatting updates and technical items listed in the 'Journal Requirements' section below.* A marked-up copy of your manuscript that highlights changes made to the original version. You should upload this as a separate file labeled 'Revised Manuscript with Track Changes'.* An unmarked version of your revised paper without tracked changes. You should upload this as a separate file labeled 'Manuscript'. If you would like to make changes to your financial disclosure, competing interests statement, or data availability statement, please make these updates within the submission form at the time of resubmission. Guidelines for resubmitting your figure files are available below the reviewer comments at the end of this letter. We look forward to receiving your revised manuscript. Kind regards, Chongzhi ZangAcademic EditorPLOS Computational Biology Jian MaSection EditorPLOS Computational Biology Feilim Mac GabhannEditor-in-ChiefPLOS Computational Biology Jason PapinEditor-in-ChiefPLOS Computational Biology  **Journal Requirements:** **Additional Editor Comments (if provided):** As you see, the reviewer raised further comments and concerns to the responses and to the revised manuscript. We suggest that you fully address their comments in a substantive revised manuscript.**Reviewers' comments:** Reviewer's Responses to Questions

**Comments to the Authors:**

Reviewer #4: None of the reviewer’s comments were adequately addressed. The authors made only slight modifications to the text and seemed to spend more effort on the rebuttal letter, arguing with the reviewer.

1. Regarding Bacarella’s paper, please carefully review ‘Additional File 4’ and consider the probability of randomly drawing 100 papers from PubMed, with multiple papers coming from the same group. This suggests it’s not truly random.

2. In Bacarella’s paper, they specifically stated, 'Studies utilizing previously published datasets, including large-scale sequencing efforts (such as TCGA) were excluded, to ensure a representative sampling of the most common experimental designs.' It’s reasonable to exclude large consortia studies and publications using large cohorts of data for their own purpose. However, you cannot conclude that most cancer studies use fewer than six replicates.

3. Once again, comparing tumor versus normal tissue doesn’t make practical sense. When inter-group signals are much stronger than intra-group variability, it doesn’t matter whether you have 6 replicates or 100. Consider an extreme case: comparing humans and E. coli—you wouldn’t need replicates at all. You cannot use such extreme cases to imply your method works across all human studies. Again, comparing tumor to normal tissue is inappropriate.

4. In real-world research, scientists and clinicians are more focused on comparing different subtypes rather than tumor versus normal within a single cancer type.

5. The authors also misunderstood the term 'subtype.' In cancer biology, subtypes refer to groups of the same cancer sharing specific characteristics, such as the four breast cancer subtypes: Luminal A, Luminal B, HER2-positive, and triple-negative. LUSC, LUAD, and BRCA are different cancers, not subtypes. Don’t argue that LUSC and LUAD are both 'lung cancer'—they are distinct cancers. The 33 cancer types studied by the TCGA consortium are clearly listed on the NCI’s website. Please follow their definitions.

6. There are many human studies that don’t have as many samples as TCGA. At least demonstrate one or two such examples, rather than relying solely on TCGA data in the main text.

7. Even if you insist on using TCGA data, there are more meaningful ways to apply it. For example, comparing the four molecular subtypes in BRCA, or comparing patients with good vs. poor survival outcomes in triple-negative BRCA, or platinum-tolerant vs. resistant patients in triple-negative BRCA.

8. The tumor vs. normal comparison across eight cancers only demonstrates a single scenario. One scenario cannot convince people of the broader applicability of your method.

Overall, while I saw the potential of the authors' work to address the replicability issue in studies with a low number of replicates, I suggested several use cases—human, non-human, and non-cancer—to improve the generalizability of their work in last round of review. These suggestions could have been implemented within one to two months. It’s very disappointing that none of my recommendations were taken seriously, and no significant improvements were made after nearly five months.

**Have the authors made all data and (if applicable) computational code underlying the findings in their manuscript fully available?**

Reviewer #4: Yes

PLOS authors have the option to publish the peer review history of their article (what does this mean?). If published, this will include your full peer review and any attached files.

Reviewer #4: No

 **Figure resubmission:**While revising your submission, please upload your figure files to the Preflight Analysis and Conversion Engine (PACE) digital diagnostic tool, https://pacev2.apexcovantage.com/. PACE helps ensure that figures meet PLOS requirements. To use PACE, you must first register as a user. Registration is free. Then, login and navigate to the UPLOAD tab, where you will find detailed instructions on how to use the tool. If you encounter any issues or have any questions when using PACE, please email PLOS at figures@plos.org. Please note that Supporting Information files do not need this step. If there are other versions of figure files still present in your submission file inventory at resubmission, please replace them with the PACE-processed versions. 
---

## [Decision Letter · Decision Letter 3]

7 Apr 2025

Dear Author,

We are pleased to inform you that your manuscript 'Replicability of bulk RNA-Seq differential expression and enrichment analysis results for small cohort sizes' has been provisionally accepted for publication in PLOS Computational Biology.

Best regards,

Chongzhi Zang

Academic Editor

PLOS Computational Biology

Jian Ma

Section Editor

PLOS Computational Biology

Reviewer's Responses to Questions

**Comments to the Authors:**

Reviewer #4: No further comments

**Have the authors made all data and (if applicable) computational code underlying the findings in their manuscript fully available?**

Reviewer #4: Yes

PLOS authors have the option to publish the peer review history of their article (what does this mean?). If published, this will include your full peer review and any attached files.

Reviewer #4: No

---

## [Editor Report · Acceptance letter]

PCOMPBIOL-D-23-01720R3

Replicability of bulk RNA-Seq differential expression and enrichment analysis results for small cohort sizes

Dear Dr Degen,

I am pleased to inform you that your manuscript has been formally accepted for publication in PLOS Computational Biology. Your manuscript is now with our production department and you will be notified of the publication date in due course.

With kind regards,

Anita Estes
